# Current Status of Internet Gaming Disorder (IGD) in Japan: New Lifestyle-Related Disease in Children and Adolescents

**DOI:** 10.3390/jcm11154566

**Published:** 2022-08-04

**Authors:** George Imataka, Ryoichi Sakuta, Akira Maehashi, Shigemi Yoshihara

**Affiliations:** 1Department of Pediatrics, Dokkyo Medical University, Tochigi 321-0293, Japan; 2Child Development and Psychosomatic Medicine Center, Dokkyo Medical University Saitama Medical Center, Saitama 343-8555, Japan; 3Faculty of Human Sciences, Waseda University, Saitama 359-1192, Japan

**Keywords:** internet gaming disorder, lifestyle-related disease, prefrontal cortex, limbic system, amygdala

## Abstract

The World Health Organization recognizes internet gaming disorder (IGD) as a disorder that causes problems in daily life as a result of excessive interest in online games. The causes of IGD have become more apparent in recent years. Because of prolonged exposure to games, the mechanisms controlling the reward system, such as the prefrontal cortex, limbic system, and amygdala of the cerebrum, do not function properly in IGD. This mechanism is similar to that of various behavioral addictions, such as gambling addiction. IGD is particularly risky in children and adolescents because it easily causes brain dysfunction, especially in the developing brain. IGD should be regarded as a new lifestyle-related disease in younger individuals, and lifestyle modifications, including counseling and family therapy, are critical for its management.

## 1. Introduction

Animation, the Internet, and video games are becoming the primary concerns among children and young adults globally [1,2]. The majority of today’s society is made up of young adults who grew up with animated characters and Internet/video games [3,4]. Internet gaming disorder (IGD) has become a global problem in recent years, as people have become overly enthusiastic about Internet games, devoting a large amount of time and energy from their personal lives to gaming [5]. Young Asian males appear to be the most prone to IGD [6,7]. Prevalence rates are highest in Eastern Asian countries and in male adolescents aged 12 years to 20 years. A Taiwanese study showed an internet gaming disorder prevalence rate of 3.1% [8], a Korean study reported a prevalence of 5.9% [9], and a Chinese study reported a prevalence of 17% [10]. The prevalence ranged from 3.3% to 18.7% among Japanese college students (based on a broader screening for Internet addiction) [11,12]. The differences in prevalence rates can be attributed to differences in study population, cultural factors, and assessment or diagnostic criteria. However, the prevalence in Asian countries seem very high [2,13] compared to the West. The diagnostic criteria for IGD are similar to those for other addictive disorders, such as pathological gambling. Neuroimaging studies on individuals with IGD have focused on structural and functional brain changes, similar to neuroimaging studies on disorders such as substance and behavioral addictions [14]. The major concern with IGD is the dependency symptoms it causes [15]. IGD patients have large-scale functional network changes that explain their core symptoms, such as low emotional competence, cue reactivity, and desire, habitual addictive behaviors, and decreased executive control [16]. According to a recent meta-analysis, one in every three IGD patients experienced depression [17]. Japanese students tend to have a high risk of IGD and physiological stress [18]. Furthermore, more people in Asian countries are showing behaviors associated with gaming disorder and online addiction amid the coronavirus (COVID-19) pandemic [19,20].

IGD has negative effects on a person’s physical, emotional, and social welfare. Pathological gaming has negative effects on one’s mental health, academic performance, physical violence, and depressive symptoms. It is also associated with poor sleep quality and sleep-related issues, decreased interest in schoolwork, and extracurricular activities [21]. Considering growing prevalence of IGD in Asia, children and adolescents deserve more clinical attention. In this expert review, the following topics are discussed: (1) what addiction is, (2) the relationship between history of games and game addiction, (3) what IGD is, (4) approaches to game addiction, and (5) treatment of IGD based on the latest findings.

## 2. What Is an Addiction?

### 2.1. Two Kinds of Addictions: Dependence on Substances and Dependence on Behaviou

Briefly, addiction is defined as a state in which a person is unable to stop using a particular substance or engaging in a particular behavior despite negative consequences [22,23]. Disorders caused by substance use include those caused by a single instance or repeated use of substances with psychoactive properties, including certain medications. In most cases, the first use of these substances produces pleasant or appealing psychoactive effects that become rewarding and reinforcing with repeated use. Many of the substances included have the potential to cause dependence if used repeatedly. They also have the potential to be harmful to both mental and physical health. Disorders caused by addictive behaviors are identifiable and clinically significant syndromes characterized by distress or interference with personal functions and caused by repetitive rewarding behaviors other than the use of dependence-producing substances (Figure 1). Gambling disorder and gaming disorder, which can involve both online and offline behavior, are examples of behavioral addictions [24]. Both substance and non-substance-related disorders refer to compulsive indulgence in a specific behavior or type of behavior that causes harm to the person, as well as the person’s inability to moderate or manage those behaviors without treatment [25]. Behavioral addictions such as substance-related disorders exhibit similar psychological and behavioral patterns, such as craving, impaired control over behavior, tolerance, withdrawal, and a high rate of relapse [26,27,28,29]. Addiction and related behavior was added to the DSM-5 in 2013 [22]. A behavioral addiction category has been proposed within this category, which would include pathological gambling and possibly Internet addiction [30,31,32]. The DSM-5 now recognizes two behavioral addictions: gambling addiction and gaming disorder. The DSM-5 Section 3 includes IGD as a condition, but Internet addiction is not included, which needs more clinical research and experience before it can be considered for inclusion in the main book as a formal disorder [33].

### 2.2. IGD in DSM-5 and ICD-11

Internet gaming disorder is a psychiatric disorder characterized by a persistent and recurring interest in video games that significantly impairs daily, professional, or educational activities [33,34]. IGD was placed in the section of substance-related and addictive disorders, together with gambling disorder, “reflecting evidence that gambling behaviors activate reward systems similar to those activated by drugs of abuse and produce some behavioral symptoms that appear comparable to those produced by the substance use disorders”. It is the only online behavioral addiction that is well recognized in both DSM-5 and ICD-11, in addition to gambling [33].

The symptoms of IGD resemble other Internet addictions and include repetition: seeking stronger stimulation, trying to stop but not being able to, and being unable to get it out of one’s head at all times. Initially, there is no difficulty; however, over time, these symptoms become exceedingly challenging for a person’s social life [34]. The DSM-5 states that five out of the nine diagnostic criteria (preoccupation or obsession, withdrawal, tolerance, loss of control, loss of interest, continued overuse, deceiving, escaping negative feelings, and functional impairment) must be met within a year in order to be diagnosed with IGD [33].

In 2022, the World Health Organization (WHO) included the category of gaming disorder in ICD-11. To be officially diagnosed with gaming disorder (“digital gaming” or “video gaming”), a patient must exhibit three symptoms: impaired control over gaming, increasing priority given to gaming over other activities to the extent that gaming takes precedence over other interests and daily activities, and continuation or escalation of gaming despite the occurrence of negative consequences [24]. The WHO’s distinction between online and offline gaming modes is significant because online gamers have the highest disordered gaming scores, followed by mixed gamers (those who say they prefer both online and offline gaming), and offline gamers [35]. In comparison to the DSM-5 IGD diagnosis criteria, the ICD-11 diagnosis criteria emphasize functional impairment [36]. A recent survey involving five Mexican universities showed that the demographics, comorbid mental disorders, service use, and impairment variables of DSM-5 cases detected and undetected by ICD-11 criteria were similar; however, cases detected by ICD-11 had severe symptoms and had a probable drug dependence compared to undetected DSM-5 cases [37].

## 3. Children and Adolescents Are Particularly at Risk for Onset of IGD

Internet game addiction is a major concern for Asia’s three largest economies, South Korea, China, and Japan. The reasons for its ubiquity in the region are complex. There are several factors attributed to the rising rate of gaming addiction. These countries have ease of access to a wide range of games, as well as computers and the Internet. The presence of 24 h video game cafés with fast Internet speeds exacerbates the gaming addiction problem.

The enormous social loss caused by game addiction has become a problem over the past decade necessitating that all stakeholders take countermeasures. In South Korea, the government has seriously considered the fact that many young people are wasting substantial time gaming. The Cinderella Act is a youth protection law, prohibiting youth under the age of 16 years from playing online games from midnight to 6:00 a.m. The system forcibly disconnects the online connection at 12:00 a.m. [38]. In South Korea, the law was enacted to counter suicides and Internet crimes, with an estimated one million “netgeeks” due to IGD.

There is currently little significant research evidence that playing games for a set period of time is associated with disorganized behavior. However, a number of negative consequences of excessive gaming enthusiasm in childhood have been identified. China issued a notice on the Prevention of Underage Online Game Addiction in 2019. Logging into online games from 10:00 p.m. to 8:00 a.m. the next morning is not allowed, and online games are shut down after 3 h on weekends and holidays and 1.5 h on weekdays [39]. In addition, billing is not allowed for those under 8 years old, with a monthly limit of about 30 USD for those between 8 and 15 years old, and about 60 USD for those between 16 and 17 years old. In China, the government is taking the lead in creating an environment that prevents gaming dependency among people. Although there is a tendency in Japan to consider gambling addiction as a series of spectrum-like conditions as IGD, Japan is lagging behind in government-led measures regarding Internet gaming addiction.

### 3.1. Age-Related Development of Neurocognition: Scammon’s Developmental Curve

In the environmental adjustment of IGD, keeping smartphones and computers away from the surroundings is important. The Japan Pediatric Association and the Society of Obstetricians and Gynecologists are working to educate people not to let smartphones get a grip on their children. The American Academy of Pediatrics has recommended that parents limit screen time for children aged 2 years or younger, when the brain is particularly malleable, due to concerns that a young, developing brain may be particularly sensitive to chronic exposure to computers, smartphones, tablets, or televisions [40]. According to Scammon’s developmental curve, where the age of 20 is 100% of the human organ-specific developmental level, the nervous system is 50% formed at age 2, 80% formed at age 4–5, and 90% formed at age 8–9 [41]. Excessive gaming and digital stimulation of the developing brain has a negative impact on subsequent brain development owing to specific neurodevelopmental plasticity in adolescents. It has been reported that the volume of the gray matter of the brain, including the prefrontal cortex, in the brains of younger individuals with IGD is smaller than the average for the same sex and age [42]. The prefrontal cortex, along with the limbic system of the brain, regulates the emotions of desire and anger, and a small portion of the prefrontal cortex reduces the ability to reason and suppress emotions. Gaming exposure at younger ages causes changes in brain morphology [6,43,44,45].

### 3.2. Online Games Are Easy to Depend on and Are Cleverly Designed

IGD disrupts the rhythm of life due to the huge amount of time spent, resulting in sleep deprivation and reward system deficiency. This reduces satisfaction and leads to a constant state of dissatisfaction. A recent study demonstrated that adolescent students and their siblings had mutual impacts of IGD on psychological health and sleep [46]. In addition, many games and Internet sites are designed as candy and whips to keep young people returning to the same sites. Common symptoms of IGD include day and night reversals (changing the sleep–wake cycle), domestic violence, destruction of property, staying indoors, not eating, not being able to get up in the morning, persistent absenteeism, and poor grades [6,43]. IGD has a greater connection with achievement motivation compared to social or immersion motivation. The desire to explore the virtual world may not be genuine motivation, but rather a means of escaping from various real-life problems [47]. In several games, people play in teams where each individual plays the role of playing in a battle game to survive against the enemies. Teams gather in a virtual space at night after school or work, and, each time they win a game, their ranking increases. Acknowledging the relevance of socializing and the relationship motivation subcomponents to IGD and its management is essential [47]. Online games not only allow players to hone their skills but also exposes them to the use of violent weapons online and increases the risk of children’s dangerous behavior around weapons such as firearms [48]. Game companies have smartly created a system that makes it easy for people to be dependent on their games, increasing their profits in the process. In other words, the longer young people spend time gaming, the greater the profit for the companies. Excessive game playing can lead to a lack of exercise and even sleep, making it easier for children and adolescents whose central nervous system is still immature in development to be more detrimental to their health than adults.

## 4. Relationship between the History of Games and Game Dependence

### 4.1. History of Games: Ancient to Modern

The history of games dates back to BC times (Figure 2). Humans have been fascinated with games since ancient times, and games have changed and developed worldwide along with human history. However, the history of games is connected to the history of gambling, betting, exposition, and addiction. From Roman emperors to Marie Antoinette, there are many historical anecdotes of people who lavishly gambled in games [49,50].

### 4.2. The Spread of Arcade Games and the Problem of Slot Machines, Pinball, and Pachinko

In 1927, David Gottlieb created an arcade game in Chicago that became the prototype for “pinball”. During the post-prohibition era in the United States, many pinball machines were installed in arcades and bars. In the 1930s, the pinball was labeled as a gambling machine, and the movement of citizens to ban pinball grew stronger. Finally, in 1936–1937, the Chicago Pinball Prohibition Act was enacted. Historically, this was the first law to prohibit gambling game machines. At the same time, pachinko parlors appeared in Japan. Pachinko is a Japanese mechanical game that is played as a recreational arcade game and much more frequently as a gambling device, occupying a niche in Japanese gambling similar to that of the slot machine in Western gambling as a low-stakes, low-strategy kind of gambling (Figure 3). As soon as slot machines, pinball, and pachinko were introduced to the Japanese market, gambling was seen as a problem.

### 4.3. Space Invaders and Regulations against Underage Game Dependence

In June 1978, Taito Corporation released the table-integrated arcade game “Space Invaders”, which was the first fixed shooter, and it laid the groundwork for the shoot ‘em up genre (Figure 4a,b). The goal is to earn as many points as possible and defeat wave after wave of descending aliens with a horizontally moving laser. Considered one of the most influential video games of all time, Space Invaders contributed to the growth of the video game industry from a novelty to a global industry, ushering in the golden age of arcade video games [51]. The game became the biggest hit for an arcade-type game. However, it caused the problem of underage gaming dependency in Japan. Within a year and a half of its release, approximately 500,000 units had been shipped in Japan. Invader cafés, which were coffee shops that turned into game centers, appeared in towns. The long waiting lists for games at Invader cafés received news coverage. Eventually, illegal Invader machines that could be played at low prices were installed in candy stores where children congregated, and adolescents flocked to these places with cash. Thus, long waiting lists for Invader games and the occupation of game machines became a problem. In due course, Invader cafés and game centers attracted adolescent delinquents. Police and school groups investigated the situation, and just 1 year after the announcement of Invader, the All-Japan Amusement Park Association declared a voluntary restraint on Invader games. In 1985, the government enforced the Entertainment Business Law prohibiting anyone under the age of 18 from entering game arcades after 10 p.m. (At present, anyone under the age of 16 is not allowed to enter game arcades in Japan after 10 p.m.). Because of the Invaders boom’s unique cultural context in Japan, the Space Invaders boom and IGD have been explored collectively. As interest in the issues of Internet and game addiction grew in Japan, the Japanese government attempted to address both issues as a series of IGDs, with specialists such as pediatric neurologists and psychiatrists.

## 5. What Is Game Addiction?

### 5.1. Background of the Invader Game Craze: Reward System and Prefrontal Cortex

Why did the industry forgo Invader games after only 1 year, despite their strong sales? Why did the Japanese government propose to ban it? The reason for the popularity of the game was that people vs. machines, slot machines, and pachinko machines were regulated by law, whereas Invaders was accessible without age restrictions. However, this led to a sharp increase in gaming disorders among children. Although there is still insufficient evidence that children’s developing brains are vulnerable to games, it was observed that children’s developing brains are vulnerable to gaming [52]. An increasing number of Japanese children are dropping out of school as a result of their excessive reliance on video games, the Internet, and social networking services. In other words, the fact that children are more highly prone to game addiction than adults has been proven by social phenomena.

IGD is a typical example of machine dependency, and the mechanism of IGD involves the release of dopamine, a pleasure substance in the brain, which activates the “reward system”, a sensor that provides a sense of accomplishment and euphoria. It was also discovered that the cause of IGD is an abnormality in the functioning of the brain that reduces the functioning of the prefrontal cortex, which controls human reasoning [43,53,54,55,56].

### 5.2. Game Addiction Is Similar to Other Addictions

Screen-based media use (including Internet-related addictive behaviors) is associated with a less efficient cognitive control system in adolescence. Adolescents with IGD more frequently exhibit decreased short- and medium-range connectivity among the subcortical, frontal, and parietal regions involved in attentional and control networks, in line with other behavioral addictions [57,58,59]. IGD is linked to functional and structural neural changes in the fronto-striatal and fronto-cingulate regions [43]. Fronto-striatal dysfunction is also thought to promote a compulsive use of the Internet and screen devices in general. It has been reported that adolescents who play action video games more frequently struggle to maintain their attention over time [58]. Adolescents typically exhibit more reward anticipation brain activity than adults do. They are more susceptible to the intense rewards that are provided by gaming activities (Figure 5). Excessive gaming on a regular basis can reinforce adolescents’ propensity to seek out instant gratification, which they value higher [58]. Several studies have tried to establish the dynamics involved in IGD etiology. Paulus et al. [6] described that IGD is a highly complex phenomenon, with interplay of psychological, sociological, and neurobiological factors. The authors proposed an integrated IGD model depicting complex interaction between serval internal and external factors. Structural brain deficits in prefrontal cortex, amygdala, connectivity, dopaminergic/serotonergic system, and neurobiological deficits (self-regulation and decision making, dysregulation of mood, and reward systems) are implicated in IGD. Among external factors, poor parental care, inadequate social skills, and game-related factors such as rewards can increase the risk of IGD. Adolescents with IGD have similar traits to those with substance dependence, including higher impulsivity, greater propensity to make risky decisions, less capacity to defer gratification, and impaired ability to assess risk. Schettler et al. [60] reported that adolescents with problematic gaming have altered brains that show deficits in executive functioning (including working memory and attention), emotion management, and reward processing, as well as cognition, which includes decision making. Adolescents with IGD seem to exhibit a stronger imbalance between cognitive control (including fronto-parietal areas) and the affective system (including subcortical and limbic structures) than non-IGDs [42]. Lower gray-matter density in the dorsolateral prefrontal cortex was linked to more severe IGD symptoms, more depression, more lifelong gaming, more impulsivity, and more time spent gaming. Previous neuroimaging studies revealed that IGD and addiction may have comparable neurobiological pathways, such as aberrant fronto-striatal networks that are important in reward processing and cognitive regulation [14]. Given that the DLPFC is essential to the top-down control system that manages cognition and behavior, its involvement in IGD is not unexpected. It can be assumed that the prefrontal dopaminergic system partially controls impulsivity, cravings for online gaming, and poor mood [42,43,52]. Furthermore, in response to punishment, IGDs were found to use an avoidance system, indicating a distinct pattern of reward processing among people with IGD [61]. Recent studies suggest that IGD individuals may have greater appetites for video games than for more fundamental rewards such as food, and they subjectively attach greater importance to gaming as the primary reward. When compared to food-related cues, gaming-related cues induced stronger functional connectivity in precuneus–caudate relationships in IGD individuals [62]. Patients with IGD demonstrated considerable bilateral hyperactivation in the precuneus and cingulate, as well as significant bilateral hypoactivation in the insula, during cue reactivity tasks, but there were no alterations in the striatum. Patients with IGD showed significant hypoactivation in the left inferior frontal gyrus and hyperactivation in the right superior temporal gyrus, bilateral precuneus, bilateral cingulate, and insula during executive control tasks. IGD patients showed significant hypoactivation in the left superior frontal gyrus, left inferior frontal gyrus, and right precentral gyrus during risky decision-making paradigms, as well as significant hyperactivation in the left striatum, right inferior frontal gyrus, and insula [63]. Some of these changes are linked to aspects of addiction or to changes in the brain’s motor, sensory, and cognitive functions [55].

### 5.3. The Need for Treatment of IGD

Individuals with IGD frequently overestimate the benefits of playing online games, where they invest a substantial amount of time. As a result, they neglect social and academic obligations, which can lead to social dysfunctions [64]. Therefore, IGD must be considered a disease. According to a survey by Japan’s Ministry of Health, Labor, and Welfare in 2017, about 97% of teenagers in Japan use the Internet on a daily basis. The results of the survey (*n* = 38,630) reported that 59.7% of users access the Internet through a smartphone, while 52.5% of them use the Internet via desktop and/or laptop computers [65]. Approximately 1.82 million males 20 years of age and older were living with Internet addiction in 2018 in Japan, almost three times the number reported in 2013 [66]. Anecdotal evidence indicates that the majority of patients with Internet addiction are primarily addicted to gaming, and male Internet users favored online gaming through smartphones [67]. Sleep disturbance, day or night reversal, inability to get up in the morning, inability to attend school, poor concentration, irritability, lack of communication, nervous anxiety, lack of exercise, obesity, and poor academic performance [6,68,69] are all symptoms of IGD in adolescents, indicating a serious concern about the dangers of excessive gaming. Moreover, over the last decade, there has been an increase in research on the mental health issues associated with IGD. Adolescents are vulnerable to a variety of mental health issues linked to excessive online gaming, including depression, social anxiety, loneliness, and attention deficit hyperactivity disorder (ADHD) [69]. The degree of depressive symptoms and physiological resilience were found to be positively correlated in a study of large Chinese secondary school students, suggesting that psychological resilience training may be helpful as an intervention for IGD [70].

## 6. Approach to Game Addiction and Treatment

### 6.1. The Pathology of Game Addiction Is Psychological Dependence

Physical dependence requires treatment for withdrawal symptoms when the drug is discontinued. Process dependence, on the other hand, is mental dependence. Therefore, IGD is expected to have a number of clinical effects through environmental adjustment, counseling, and cognitive behavioral therapy [6,44,45].

### 6.2. Diagnosis of Game Dependence

IGD is difficult to recognize until it becomes serious, and it tends to be delayed until later. Therefore, the first step in dealing with IGD is for the patient and family to self-recognize IGD. Outpatient diagnosis of IGD and counseling of the patient and family are important [6]. For this purpose, first, a therapeutic target should be approached from family therapy. Data on management are available for younger patients, but not for children. Few cases have been reported emphasizing early treatment in children in order to avoid a longer and more intensive approach when the child progresses to higher education. In children, the clinical approach should include assessment, rapport building, parent education, child/teenager education, child/teenager motivational interviewing, and individual therapy based on psychological symptoms.

### 6.3. Toward Less Games, Not More, and Zero-Game Day

The next thing to consider is the process via which gaming can be dealt with. It might be a beneficial idea to dispose of the games; however, this should be evaluated on a case-to-case basis. First and foremost, people with IGD have trouble managing their reasoning; as such, if games are forced away from them, they are at risk of impulsive behaviors, such as violence and running away from home [6,43]. Since IGD is basically a psychological dependence [71,72], the next step is to counsel the person on how to deal with games once they are aware of the dependence. If it is difficult to stop playing games, the aim is to reduce the number of games played. Working not only with the individual but also with the family as much as possible is important. In several cases, the inability to escape IGD may be caused by members of the family [73].

### 6.4. IGD Initiatives in Japan

According to a survey of Youth Internet Environment 2018, Cabinet Office Japan, 95.9% of senior-high-school students, 58.1% of junior-high-school students, and 30% of elementary-school students have their own smartphones [74]. Junior- and senior-high-school students use their smartphones for greater than 4 h a day on average, and 10–20% of Japanese junior- and senior-high-school students are already suffering from IGD. Therefore, it is important for families and schools to discuss gaming as a collective. Additionally, social regulations and measures are necessary. For example, the Japan Pediatric Association has issued a call for “Smartphone Time, What Am I Ushering in?” Kanagawa Prefecture has uploaded an animated YouTube video on its website and has over 30,000 registered users. Kagawa Prefecture enacted Japan’s first gaming regulation ordinance on 1 April 2020. The ordinance states that games must be used for 60 min and 90 min on weekends and holidays, and that smartphones must be used until 9:00 p.m. for junior-high-school students and younger, and until 10:00 p.m. for all others, except for those used for study purposes [75]. Although the ordinance has no penalties, it has been criticized by libertarians. However, given the fact that the WHO has recognized IGD as a disease and the government has not set any laws, several people in Japan are of the opinion that the ordinance of Kagawa Prefecture should be enforced.

## 7. Treatment of IGD

### 7.1. Actual Treatment of IGD

There is a scarcity of literature on various IGD treatment methods. No standard clinical treatment protocol is currently available, and treatment techniques are typically derived from those used to treat substance use or gambling disorders. In the literature, individual, group, and family psychotherapeutic interventions, pharmacotherapy, and addiction clinics with multimodal treatment programs have all been mentioned in adolescents [76,77,78]. Importance is placed on making the person self-aware regarding the problems in their life and what they may face in the future due to gaming. In addition, the medical staff should support the individual and the family in successfully dealing with gaming and smartphones. The multidimensional family therapy (MDFT) approach seems to be a viable option for IGD. MDFT exemplifies a practical, adaptable, and widely transportable approach. The MDFT approach is being applied to IGD cases in two ongoing studies in France and Switzerland [78,79].

Psychological counseling and discussions with other children are also important. Cognitive behavioral therapy, in which the patient and the family live in an environment without smartphones, such as camping or hospitalization, is also important [80]. It is also necessary to visualize and self-manage the time spent gaming and on smartphones to return to the same environment after discharge [81]. To improve IGD, a multifaceted approach involving not only patients, families, and doctors, but also companies, governments, and administrators is required, and preventive measures are currently being considered in Asia from a variety of perspectives.

Medications such as bupropion, escitalopram, or methylphenidate have been used to treat IGD and/or comorbid psychopathology (e.g., depression, anxiety, and ADHD) [76]. A recent meta-analysis involving 5601 children and young adults with Internet addiction/IGD indicated that pharmacotherapy combined with CBT or multilevel counseling might be an effective therapeutic strategy for youth with gaming disorder [82].

### 7.2. Proposition: IGD Is a New Lifestyle-Related Disease in Children

IGD has emerged as a new behavioral addiction that appeals to the younger generation despite risks to their physical, mental, social, or financial wellbeing. With the rapid proliferation of online games, there is a risk that IGD will promote sedentary behavior, irregular eating patterns, and other unhealthy lifestyle choices in children [21]. Furthermore, several studies have found that young adolescents are at risk of substance abuse [83]. New devices are being released worldwide, and the next boom is occurring repeatedly. However, the negative effects of these devices are not observed until a little later. Today, the population of people who enjoy the Internet and games has grown too large to conclude that they are uniformly inadequate.

The concern is that IGD is now a lifestyle disease for children. The basis of measures against lifestyle-related diseases is to raise awareness in society to prevent the onset of the condition.

## 8. Conclusions

Despite being identified as a separate entity by DSM-5, the true scope of the IGD is yet to be determined and recognized as a valid public health concern. Foremost, there is a diagnostic ambiguity involving Internet addiction and IGD that must be resolved. Most importantly, research into the management of IGD is in its infancy and needs to accelerate. There is an urgent need to raise awareness and formulate effective clinical guidelines for the management of IGD. On the other hand, there has been extensive research on game theory and gamification, facilitated by advances in information technology in society. It is hoped that this discussion will not hinder the progress of technology so that the technology of game development can benefit humanity.

## Figures and Tables

**Figure 1 jcm-11-04566-f001:**
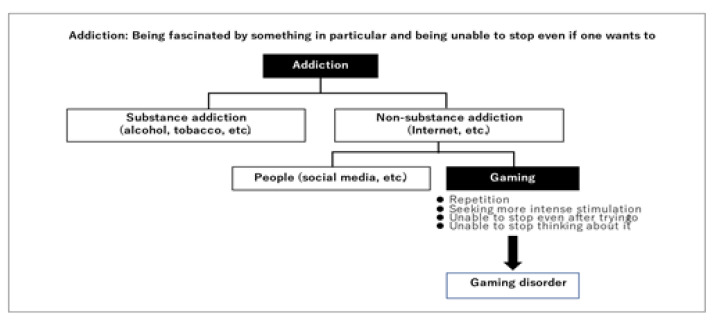
Two kinds of addiction: what is gaming disorder?

**Figure 2 jcm-11-04566-f002:**
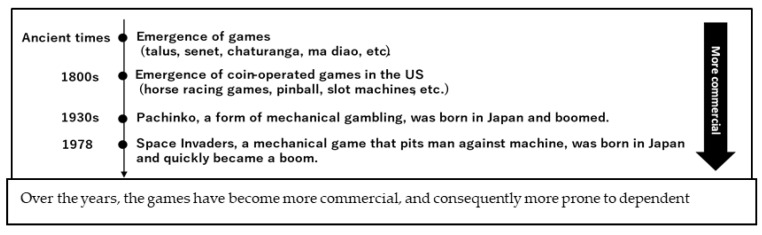
History of gaming: From “person vs. person” to “person vs. machine”.

**Figure 3 jcm-11-04566-f003:**
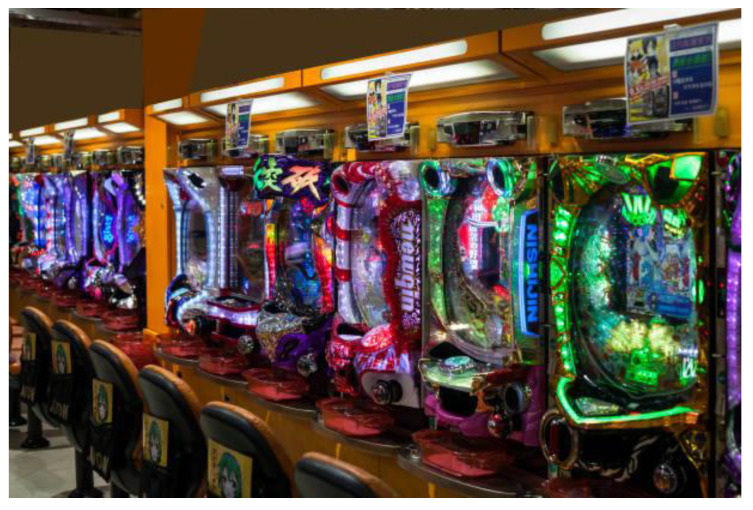
Pachinko parlor.

**Figure 4 jcm-11-04566-f004:**
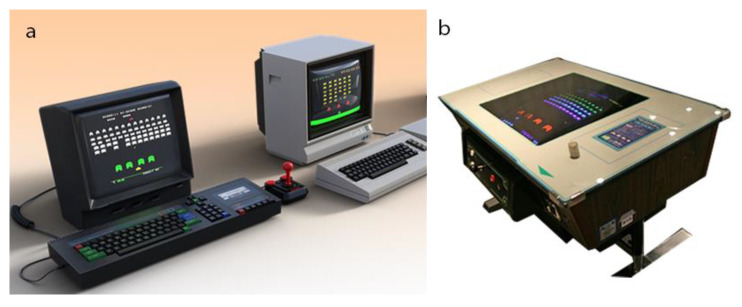
Space invaders game: (**a**) computer old retro version; (**b**) table-top version game.

**Figure 5 jcm-11-04566-f005:**
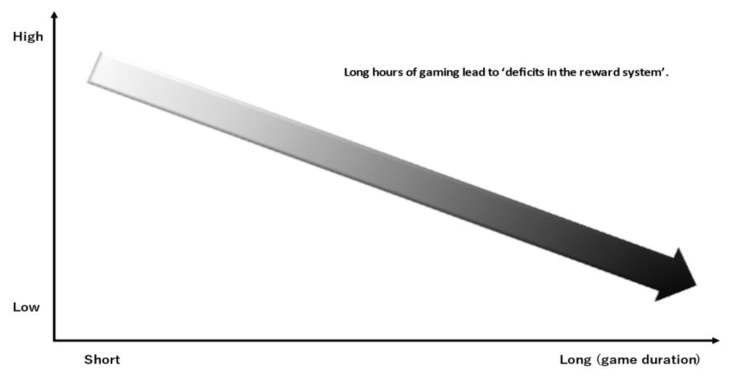
The reward system and the prefrontal cortex (level of satisfaction/activity of the reward system).

## Data Availability

Not applicable.

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
