# Peer review of "Current Status of Internet Gaming Disorder (IGD) in Japan: New Lifestyle-Related Disease in Children and Adolescents"

_jcm, 2022, doi:10.3390/jcm11154566_

Round 1

Reviewer 1 Report

This is an interesting review on an important topic. Compared to the first submission this re-submission is distinctly improved. The state of the art related to the topic is clearly elborated. In general the manuscript is well written.

Only some minor corrections are requested.

line (l) 205, please change proposed to propose

l 311 please change means to mean

l 311/l 312 - Why the reference Wang 2022 is not coded. Is it given in the reference list?

l 366 - pleas rephrase  "seriously consideration"

reference 36 - please change 19918 to 199, 18

Author Response

This is an interesting review on an important topic. Compared to the first submission this re-submission is distinctly improved. The state of the art related to the topic is clearly elborated. In general the manuscript is well written.

Response: Thank you for your comments. 

Line (l) 205, please change proposed to propose

Response: Thank you for the suggestion. We have now revised it.

l 311/l 312 - Why the reference Wang 2022 is not coded. Is it given in the reference list?

Response: Thank you for the suggestion. We have now revised it.

l 366 - pleas rephrase  "seriously consideration"

Response: Thank you for the suggestion. We have now revised it.

reference 36 - please change 19918 to 199, 18

Response: Thank you for the suggestion. We have now revised it.

Reviewer 2 Report

Dear Editor,

Many thanks for the opportunity to revise the article: “Current status of Internet Gaming Disorder (IGD) in Japan: New lifestyle related disease in children”, which examines the prevalence, history, neural correlates, consequences, and treatment of Gaming disorder in Japan and Asian countries in children.

Although the topic is very interesting, I think the paper does not fit for publication in the current format. The review is narrative which is okay, but I think it lacks a clear structure and focus. The authors started to differentiate substance vs behavioural addictions. However, I think that they should develop this part more, report more clearly the criteria according to dsm5 and icd-11, highlight differences, define better why children and not adolescents, which age range they want to focus on, etc. I found some parts related to the definition later in the manuscript (eg section 5.1) so I suggest the authors start very clearly with the definitions, then report prevalence rates in young people (define children age range or also include adolescents) and I would recommend to either expand the focus on Asian countries (since they later Korea and China) or just stick to Japan, but report why you want to focus on Japan solely, and how Japan is different from other Asian countries respect to this topic. I appreciate the effort to describe the history behind gaming disorder, but I don’t think this journal is a good outlet for such historical insight. I suggest either cutting or shortening that part. I would focus the entire paper more on young people (sometimes prevalence or studies on adults are reported and this is confusing). Also I did not catch the difference btw sections 5 and 6, since the are some overlapping points. In general, it’s difficult to read the manuscript fluently since sections are not following each other in a logical way. I suggest rearranging the manuscript as follows: Intro, definition of IGD, why it is important in Japan (or Asia)  and why focusing on young people (children/adolescents? Please define better), neural correlates in young people and comparison with other digital addictions, psychological consequences, other consequences, treatments, general considerations, conclusions and next steps. Also, I missed what you intend for lifestyle behaviour, the definition is not clear.

Also, see my other comments below:

-        Define better lifestyle related disease

-        Lines 30-31 p.1 the description of dependency is too simplistic

-        Line 41: what a “behavioural addiction is”, you should acknowledge the difference

-        Lines 50-52 p. 2 the difference btw process and behavioural addiction is not clear. Sincerely, now the WHO talks about

-        Lines 54-55 p. 2 the description of the DSM-5 classification should be stated more clearly: the Gaming disorder is present in section III (but Internet addiction is not included and it raised more controversy). I would stick to the Gaming problem, stating that it is the only online behavioural addiction that is well recognized both in DSM5 and ICD-11 in addition to gambling

-        Indeed, I would merge section 2.1 and section 2.2, focusing on gaming disorder and how it is classified in both dsm5 and icd-11, and report all the criteria in more detail.

-        You should extend the focus to Asian countries and not only Japan.

-        The historical part is quite out of the topic, I would cut or shorten section 3. Also, I do not see the connection with the focus on children.

-        Why do you focus on children and not adolescents as well? the adolescent brain is even more susceptible to rewards

-        In section 4.2 I would include more reviews of the literature on Gaming disorders and brain correlates, as well as other reviews related to Internet additions in young people (https://pubmed.ncbi.nlm.nih.gov/29633243/, https://onlinelibrary.wiley.com/doi/full/10.1111/adb.13093, https://www.sciencedirect.com/science/article/pii/S0149763416305917?casa_token=VnFLDrv6elsAAAAA:Xz8aUyAA-FUmCH9A30N3gRwUqGqwtQjn_zqWeTz2tNpKexeLkGS1ZORI7ngwMpLyU6i2p3bC, https://www.frontiersin.org/articles/10.3389/fpsyg.2021.671817/full, )

-        4.3. should focus on the treatment of young people

-        Section 5.1 should be put at the beginning when you differentiate between substance and behavioural addictions

-        You also talked about Korea and Japan, so just try to focus on Asian countries

-        Section 5 is too scattered and badly connected with section 6

Author Response

To

Robbie Xie,

Journal of Clinical Medicine, section Clinical Psychology

Subject: Submission of revised manuscript (jcm-1824665)

Dear Robbie,

We would like to thank Journal of Clinical Medicine, section Clinical Psychology for a thorough assessment of manuscript report entitled “Current status of Internet Gaming Disorder (IGD) in Japan: New lifestyle related disease in children” and for providing valuable inputs for improving its quality.

We greatly appreciate the opportunity for providing a revision. We would like to thank the reviewers for their comments on our manuscript and have amended the manuscript as suggested. We have also provided a point-by-point response to each comment raised by the reviewers.

We are hopeful that the revisions proposed will improve the chances of our case report being accepted in Journal of Clinical Medicine, section Clinical Psychology and we look forward to hearing from you.

Thank you so much for your consideration

Sincerely,

George Imataka,

Department of Pediatrics, Dokkyo Medical University,

Tochigi, Japan

Email id: geo@dokkyomed.ac.jp

CONFIDENTIAL TO AUTHORS

Reviewer #2 (Reviewer Comments to the Author):

Although the topic is very interesting, I think the paper does not fit for publication in the current format. The review is narrative which is okay, but I think it lacks a clear structure and focus. The authors started to differentiate substance vs behavioural addictions. However, I think that they should develop this part more, report more clearly the criteria according to dsm5 and icd-11, highlight differences, define better why children and not adolescents, which age range they want to focus on, etc. I found some parts related to the definition later in the manuscript (eg section 5.1) so I suggest the authors start very clearly with the definitions, then report prevalence rates in young people (define children age range or also include adolescents) and I would recommend to either expand the focus on Asian countries (since they later Korea and China) or just stick to Japan, but report why you want to focus on Japan solely, and how Japan is different from other Asian countries respect to this topic. I appreciate the effort to describe the history behind gaming disorder, but I don’t think this journal is a good outlet for such historical insight. I suggest either cutting or shortening that part. I would focus the entire paper more on young people (sometimes prevalence or studies on adults are reported and this is confusing). Also I did not catch the difference btw sections 5 and 6, since the are some overlapping points. In general, it’s difficult to read the manuscript fluently since sections are not following each other in a logical way. I suggest rearranging the manuscript as follows: Intro, definition of IGD, why it is important in Japan (or Asia)  and why focusing on young people (children/adolescents? Please define better), neural correlates in young people and comparison with other digital addictions, psychological consequences, other consequences, treatments, general considerations, conclusions and next steps. Also, I missed what you intend for lifestyle behaviour, the definition is not clear.

Response:  Thank you for your comments and suggestions. We have now revised the manuscript based on the comments. In discussing game addiction, it is important to describe the relationship between people and games from the perspective of the history of games. We believe this is one of the originalities of this paper. However, as suggested we have shortened the section. We also rearranged the manuscript as suggested but included the prevalence in the Introduction section. We have given the responses to the specific questions below.

Define better lifestyle related disease

Response: Thank you for your comment. We intend to state that online games encourage sedentary behaviour, irregular eating patterns, and other unhealthy lifestyle choices, all of which are harmful to one's health.

Lines 30-31 p.1 the description of dependency is too simplistic

Response: Thank you for your comment. We have now modified the statement.

Line 41: what a “behavioural addiction is”, you should acknowledge the difference

Response: Thank you for your comment. We have now modified the statement.

       Lines 50-52 p. 2 the difference btw process and behavioural addiction is not clear. Sincerely, now the WHO talks about

Response: Thank you for your comment. Bothe process and behavioural addictions are same. We have now revised the same.

Lines 54-55 p. 2 the description of the DSM-5 classification should be stated more clearly: the Gaming disorder is present in section III (but Internet addiction is not included and it raised more controversy). I would stick to the Gaming problem, stating that it is the only online behavioural addiction that is well recognized both in DSM5 and ICD-11 in addition to gambling

Response: Thank you for your comment. We have now revised and added more IGD statements in relation to DCM-5 and ICD-11.

Indeed, I would merge section 2.1 and section 2.2, focusing on gaming disorder and how it is classified in both dsm5 and icd-11, and report all the criteria in more detail.

Response: Thank you for your comment. We intend to keep two sections: 2.1 on addiction in general and 2.2 specific to IGD.

You should extend the focus to Asian countries and not only Japan.

Response: Thank you for your suggestion. While South Korea and China have taken these measures against online games, Japan still has many pachinko parlors and an addiction problem. The latest pachinko machines are often based on various popular Japanese animated characters, which is becoming one of the causes of pachinko addiction among young people. Furthermore, the Corona epidemic in Japan has made it impossible for addicts to play pachinko and other forms of pachinko. More and more people are coming from far away from all over Japan to play pachinko in local stores in areas where the corona epidemic is less prevalent. The news reports that this is not good for preventing the spread of the coronavirus.

Japan, which has lagged behind in pachinko addiction, has similarly lagged behind in government-led measures regarding Internet gaming addiction. While China has the Cinderella Law and South Korea has adopted measures against gaming disorders, Japan, like pachinko addiction, lags behind China and South Korea in taking measures against gaming addiction.

However, we have now included data from other Asian countries since the prevalence is high in Asia.

The historical part is quite out of the topic, I would cut or shorten section 3. Also, I do not see the connection with the focus on children.

Response: Thank you for your suggestion. We have now shortened the section related to the history of games.

Why do you focus on children and not adolescents as well? the adolescent brain is even more susceptible to rewards.

Response: Thank you for your important suggestion. We have now included information on adolescents.

In section 4.2 I would include more reviews of the literature on Gaming disorders and brain correlates, as well as other reviews related to Internet additions in young people (https://pubmed.ncbi.nlm.nih.gov/29633243/, https://onlinelibrary.wiley.com/doi/full/10.1111/adb.13093, https://www.sciencedirect.com/science/article/pii/S0149763416305917?casa_token=VnFLDrv6elsAAAAA:Xz8aUyAA-FUmCH9A30N3gRwUqGqwtQjn_zqWeTz2tNpKexeLkGS1ZORI7ngwMpLyU6i2p3bC, https://www.frontiersin.org/articles/10.3389/fpsyg.2021.671817/full, )

Response: Thank you for your suggestion and references. We have now included more information in section 4.2

4.3. should focus on the treatment of young people

Response: Thank you for your suggestion. We have now emphasized treatment for young people in the revised manuscript.

Section 5.1 should be put at the beginning when you differentiate between substance and behavioural addictions

Response: Thank you for your suggestion. We have revised the statement. However, we have partly retained the definitions for better flow and clarity.

You also talked about Korea and Japan, so just try to focus on Asian countries .

Response: Thank you for your suggestion. We have more information from other Asian countries.

Section 5 is too scattered and badly connected with section 6

Response: Thank you for your comment. We have now modified sections 5 and section 6 for better flow and clarity.

Reviewer 3 Report

Thank you for providing a manuscript regarding the important topic of Internet Gaming Addiction and proposing treatment approaches for it. 

Author Response

Thank you for providing a manuscript regarding the important topic of Internet Gaming Addiction and proposing treatment approaches for it. 

Response: Thank you for your review and comments.

Round 2

Reviewer 2 Report

Please, see my few comments in the attached file. I think that the authors did a great job in revising the manuscript and I appreciated the effort. I think it is now ready for publication (just fixed the few things about the terminology of behavioral addictions also named  Non-Substance-Related Disorders. I suggest being consistent in the labels (using just one of the two throughout the manuscript). 

Author Response

Please, see my few comments in the attached file. I think that the authors did a great job in revising the manuscript and I appreciated the effort. I think it is now ready for publication (just fixed the few things about the terminology of behavioral addictions also named  Non-Substance-Related Disorders. I suggest being consistent in the labels (using just one of the two throughout the manuscript). 

Response:  Thank you for your comments and suggestions. We have now revised the manuscript based on the comments. We have now used the term “behavioral addictions’ consistently. We also accept the suggestions in the pdf and have made necessary revisions.

This manuscript is a resubmission of an earlier submission. The following is a list of the peer review reports and author responses from that submission.

Round 1

Reviewer 1 Report

The manuscript outlines a review of Internet Gaming Disorder in Japan. The manuscript covers a lot of ground, and the use of figures was good to demonstrate points, but I have a number of major concerns about the conceptualisation of IGD and some of the claims the authors made.

The terminology is confused throughout the manuscript. IGD is not a disorder recognised in the ICD-11, Gaming Disorder is, and was approved for inclusion in 2018-9 (not 2021, or 2022, when the ICD was rolled out). Although similar, they have different diagnostic criteria. This ought to be amended.

The manuscript also makes a number of questionable conceptual claims. For instance in section 2.2 the authors consider internet addiction to be subdivided into social networking and gaming addictions. This isn't a claim that is widely supported in the literature. The claim made in Figure 2, specifically that the history of games is the history of addiction, is questionable and hyperbolic. Similarly at times the manuscript descends into social commentary (i.e. about young people becoming e-sports players or YouTube creators) that I'm not convinced is necessary when providing a review of IGD. In section 5.7 the authors also write approvingly of China's restrictions on gaming to 90 minutes a day. There is little evidence a set amount of gaming is related to disordered behaviour, and there have been other studies pointing out this is unlikely to be effective (https://akjournals.com/view/journals/2006/10/4/article-p849.xml). More notably, although the authors claim there is no treatment for IGD, there is a growing literature looking at IGD treatment that isn't covered in great detail. These points ought to be addressed.

The structure of the manuscript could be made easier to follow. The abstract does not seem representative of the content of the manuscript. The paper went from discussing addiction to IGD, then to a history of gaming in Japan, back to current IGD. This ought to be streamlined. It was also difficult to pick up the key take home message from this paper. Similarly I didn't get a strong feel for who the intended audience of this paper was. The paper didn't go into enough detail for a specialist (i.e. IGD) audience, but I think at times wasn't sufficiently focused for a non-technical audience.

There are multiple instances where the references do not match up to the points made. For instance, the third sentence talks about various age groups enjoying games and anime, but the reference is a paper on the epidemiology of internet addiction. I thought the sections on the history of gaming were interesting, but they are unreferenced.

Reviewer 2 Report

This review deals with an interesting topic – Internet Gambling Disorder (IGD) in children in Japan.

Various clarifications and corrections of the manuscript are necessary.

  1. The search strategy of this review is not desribed. In the Introduction, line (l) 34, the authors write that „the following topics will be discussed.“ Does this mean that they intend to discuss a health problem in children rather than to present the state of the art in this field?
  2. L 11 – Here IGD is described as a disorder that causes problems in daily life due to overenthusiasm for the Internet and online games. Maybe „and“ was inserted by error. Anyway this word is misleading and is at odds with other reports. APA(2013) stated that „… recent scientific reports have begun to focus on the preoccupation some people develop with certain aspects of the Internet, particularly online games. The “gamers” play compulsively, to the exclusion of other interests, and their persistent and recurrent online activity results in clinically significant impairment or distress… At this time, the criteria for this condition are limited to Internet gaming and do not include general use of the Internet, online gambling or social media.“ Another report: Internet gaming disorder (IGD) is the problematic use of computer games (whether online or offline). (Wartberg L et al, 2017). These citations are not in line with the authos‘ defintion of IGD.
  3. According to DSM-5 nine criteria exist for IGD, which should be reported in the review.
  4. L 24 - a problem whic exists in „several“ people is not big. Please rephrase and clarify how big it is.
  5. L 25 – since the development of the Internet started in the early 1990 it is unlikely that the majority of the living adults worldwide and/or in Japan has grown up with Internet games. Are there references showing this?
  6. L 40 – please provide references for the definition of addiction
  7. L 49- are all Internet-related addictions mentioned here included in DSM-5 and ICD-11?
  8. L 67 – how is „the enormous social loss caused by Internet and game addiction“ shown? Again, is it appropriate to mix game addicition wiht Internet use? Aren’t many aspects of Internet and smartphones beneficial for mankind?
  9. Are there epidemiological data on IGD in general and particularly in children worldwide and particularly in Japan?
  10. Why do you mention that David Gottlieb was a Jew? You may just report that he created this arcade game. His religion is not relevant in this context.
  11. 2 and other places – Please explain shortly Pachinko and „Space Invaders“ for readers living outside Japan.
  12. L 141 – Does industry play games? Please rephrase.
  13. L 146 – „mankind“ appears to be overstated since this problem appears to be restricted to Japan. Please provide references for 4.1, 1st §.
  14. L 169/170 – „According..“ – incomplete sentence, please rephrase.
  15. L 192 – are children at a age of two years already mature for digital devices? Who has shown this?
  16. L 215 – What is meant with „stonger weapons“?
  17. L 221 - „that makes it easy for children to be addicted“ – please explain your hypothesis.
  18. L 231 – „lousy“- please rephrase.
  19. L 233- suicide would be a very serious consequence of IGD. Please elaborate in more detail.
  20. L 239 „lies within the family…“ – please gibe references showing this.
  21. L 284- 285- The prescription of sleeping pills and anxiolytics in IGD appears problematic in view of the risk of addiction due to these substances. Prescribing medications for ADHD and antipsychotics for IGD also seems questionable and should be critically discussed by the authors. Are there any studies that have shown the benefit of these substances in IGD?
  22. L 293-295 – please provide references showing the high income of these people.

Round 2

Reviewer 1 Report

I appreciate the authors have extensively amended the manuscript in response to the first round of reviews. The revised manuscript is much better written.. However, I still have a number of concerns that I do not feel have been fully addressed.

  • More generally, while I like that the authors have tried to bring several ideas together (e.g. the Space Invaders Craze and IGD), I do not feel that the evidence presented is of as strong a connection between these as the authors purport, which is where my concern regarding the the flow of the first manuscript arose. For instance, the authors state that scientists rushed to understand gaming disorder, but as a whole there was little serious research into gaming addictions until around 20 years after Space Invaders was developed and popularised. As far as I can tell (in large part based on section 3.5), the point made in 4.1 about invader games being banned is erroneous? The reference for the point concerning "it was observed that children's developing brains are vulnerable to gaming.", Ross et al 1982, is a case study of three adult men aged 25 to 35. The link between these games and other issues (such as child mental health) do not appear to be as strong as the authors state.

  • In section 4.3, the authors cite 2017 survey data suggesting that 12.4% of junior high, and and 16% of high school students use smartphones. In section 5.6, the authors cite 2018 survey data highlighting that 95.9% of senior high and 58.1% of junior high school students. This discrepancy is notable and ought to be explained.

  • While I appreciate the point regarding shift from man vs man, to man vs. machine games, the transition from slot machines to pinball and then arcade games misses that there is the inclusion of a skilful element with the latter games that distinguishes it from gambling.
  • There are still widespread terminological confusions across the manuscript. The WHO doesn't recognise IGD as a condition. The confusion is particularly concerning as most of the manuscript focuses on offline gaming. There are a number of passages e.g. Section 5.6 which are mostly about smartphone usage, not gaming. Section 2.1 - the DSM does not consider IGD as a distinct condition at this point, and should be clarified as such.
  • There's a textbox at the bottom of page 3 that has an unfinished sentence.

Author Response

I appreciate the authors have extensively amended the manuscript in response to the first round of reviews. The revised manuscript is much better written. However, I still have a number of concerns that I do not feel have been fully addressed.

Response:  We are glad for your remarks. We have now revised the manuscript based on the comments.

Comment 1. More generally, while I like that the authors have tried to bring several ideas together (e.g. the Space Invaders Craze and IGD), I do not feel that the evidence presented is of as strong a connection between these as the authors purport, which is where my concern regarding the the flow of the first manuscript arose. For instance, the authors state that scientists rushed to understand gaming disorder, but as a whole there was little serious research into gaming addictions until around 20 years after Space Invaders was developed and popularised. As far as I can tell (in large part based on section 3.5), the point made in 4.1 about invader games being banned is erroneous? The reference for the point concerning "it was observed that children's developing brains are vulnerable to gaming.", Ross et al 1982, is a case study of three adult men aged 25 to 35. The link between these games and other issues (such as child mental health) do not appear to be as strong as the authors state.

Response: We thank you for the comment. More generally speaking, we have discussed the Space Invaders boom and IGD collectively in section 3.5, for example, because of the unique cultural context of the Invaders boom in Japan. This is a point for which detailed thesis evidence is scarce. Until about 20 years after Space Invaders was developed and popularized, there was little serious research on gaming addiction. However, it is clear from the sequence of events that the generation of game addicts who were part of the Invader boom still foreshadows the current situation in Japan, where not only children but also adults are addicted to games, even though more than 20 years have passed and they are now the parents of adults.

As interest in the issues of Internet and game addiction in Japan grew, the Japanese government sought to address both problems as a series of issues as IGDs, with specialists such as pediatric neurologists and psychiatrists.

We have now added two new figures: Figure 4 on the Pachinko parlor and Figure 5 on the Space invaders game to give readers all over the world a better understanding of the games, which were originally developed in Japan.

As far as section 3.5 is concerned, section 4.1

Invader games were not actually banned, but there was a history of voluntary restraint due to social pressure from elementary school parent-teacher associations and other groups until the manufacturers voluntarily regulated them. Although there is still insufficient evidence that children's developing brains have been observed to be vulnerable to games, Ross et al 1982 in reference, for example, is not without possibility when looking at a case study of three adult males aged 25 to 35. In fact, an increasing number of children in Japanese society are dropping out of school due to excessive dependence on video games, the Internet, and social networking services, and many adults are unable to concentrate on their work. These IGD problems are unmapped by socially unhealthy lifestyles, and cognitive-behavioral therapy is being linked to lifestyle modification for public health and mental health reasons. In light of this fact, a multifaceted approach that includes not only patients, families, and doctors, but also companies, governments, and administrators is needed to improve IGD, and preventive measures are currently being considered in Japan from a variety of perspectives.

Comment 2.  In section 4.3, the authors cite 2017 survey data suggesting that 12.4% of junior high, and 16% of high school students use smartphones. In section 5.6, the authors cite 2018 survey data highlighting that 95.9% of senior high and 58.1% of junior high school students. This discrepancy is notable and ought to be explained.

Response: Please accept my apologies for any confusion. We have now corrected the information below

According to a survey by Japan’s Ministry of Health, Labour and Welfare in 2017, about 97% of teenagers in Japan use the internet on a daily basis., 12.4% of junior high school students, and 16.0% of high school students use smartphones. The results of the survey (n = 38,630) reported that 59.7% of users access the internet through a smartphone, while 52.5% of them use the internet via desktop and/or laptop computers.42 This figure is 1.8 times higher than that reported by the same survey conducted in 2012, and it was estimated that approximately 10–20% of middle and high school students had IGD.  Approximately 1.82 million males 20 years of age and older, were living with internet addiction in 2018 in Japan, almost three times the number reported in 2013.43 Anecdotal evidence indicates the majority of patients with internet addiction are primarily addicted to gaming and male internet users favored online gaming through smartphones.44

Comment 3. While I appreciate the point regarding shift from man vs man, to man vs. machine games, the transition from slot machines to pinball and then arcade games misses that there is the inclusion of a skilful element with the latter games that distinguishes it from gambling.

Response: We thank you for the comment. Section 3.2 chapter outlines the transition from human-to-human to human-to-machine gaming. As the product shifted from human-to-human to human-to-machine, the commercial component grew and so did the size of the gaming market. It is important to note, however, that in the transition from slot machines to pinball and then to arcade games, for example, a gambling element was added to these games in addition to the commercial element. The shift from human-to-human to human-to-machine games alone has increased the population of a new generation of game enthusiasts, and the addition of a financial gambling element to the mix, as well as the development of such venues for adults, especially in bars, makes the process of adding a gambling addictive element interesting when considering the mechanism of addiction. It is very interesting to consider the mechanism of addiction. Perhaps the involvement of financial interests may suggest the possibility that various products can lead people to become addicted. The shift of products from human-to-human to human-to-machine games and their commercial expansion, along with the addition of a financial gambling element, may be considered a commercial culture model that incorporates marketing and consumer psychology.

Comment 3. There are still widespread terminological confusions across the manuscript. The WHO doesn't recognise IGD as a condition. The confusion is particularly concerning as most of the manuscript focuses on offline gaming. There are a number of passages e.g. Section 5.6 which are mostly about smartphone usage, not gaming. Section 2.1 - the DSM does not consider IGD as a distinct condition at this point, and should be clarified as such.

Response: Please accept my apologies for any confusion. We have now revised the statement as below

The DSM-5 now recognizes two behavioral addictions: gambling addiction and internet gaming disorder. There is a tendency in Japan to consider both as a series of spectrum-like conditions as IGD; however, the DSM does not consider IGD as a single, independent condition.

In section, 4.3 we have removed the reference to WHO. Now the statement reads as below

The root cause of both invasive addiction and IGD, is a disease caused by abnormal brain balance.

Comment 4: There's a textbox at the bottom of page 3 that has an unfinished sentence.

Response: We thank you for the suggestion. We have now revised the statement.

Reviewer 2 Report

The manuscript was improved distinctly. I am satisfied with the revision. No further changes are necessary. 

Author Response

Response: Thank you for your remarks and comments. We are pleased that our revisions met your expectations.